# Learning What's Real: Disentangling Signal and Measurement Artifacts in Multi-Sensor Data, with Applications to Astrophysics

**Pablo Mercader-Perez**
Massachusetts Institute of Technology
pablomer@mit.edu

**Carolina Cuesta-Lazaro**
Flatiron Institute, Simons Foundation
Institute for Advanced Studies
cuestalz@mit.edu

**Daniel Muthukrishna**
Massachusetts Institute of Technology
AstroAI, CfA | Harvard & Smithsonian
danmuth@mit.edu

**Jeroen Audenaert**
Massachusetts Institute of Technology
jeroena@mit.edu

**V. Ashley Villar**
Harvard University
The NSF IAIFI
ashleyvillar@cfa.harvard.edu

**David W. Hogg**
New York University
Flatiron Institute, Simons Foundation
david.hogg@nyu.edu

**Marc Huertas-Company**
Instituto de Astrofísica de Canarias
mhuertas@iac.es

**William T. Freeman**
Massachusetts Institute of Technology
billf@mit.edu

## Abstract

Data collected from the physical world is always a combination of multiple sources: an underlying signal from the physical process of interest and a signal from measurement-dependent artifacts from the sensor or instrument. This secondary signal acts as a confounding factor, limiting our ability to extract information about the physics underlying the phenomena we observe. Furthermore, it complicates the combination of observations in heterogeneous or multi-instrument settings. We propose a deep learning framework that leverages overlapping observations, a dual-encoder architecture, and a counterfactual generation objective to disentangle these factors of variation. The resulting representations explicitly separate intrinsic signals from sensor-specific distortions and noise, and can be used for counterfactual view generation, parameter inference unconfounded by measurement distortions, and instrument-independent similarity search. We demonstrate the effectiveness of our approach on astrophysical galaxy images from the DESI Legacy Imaging Survey (Legacy) and the Hyper Suprime-Cam (HSC) Survey as a representative multi-instrument setting. This framework provides a general recipe for scientific and multi-modal self-supervised pretraining: construct training pairs from overlapping observations of the same physical system, treat sensor- or modality-specific effects as augmentations, and learn invariant representations through counterfactual generation.

## 1 Introduction

Observational data collected from the physical world can be viewed as the result of a causal process. First, an event of interest occurs, generating a physical signal. For example, a distant pulsating star emits electromagnetic waves, a person speaking creates pressure waves in the air, and a beating heart produces pressure waves that propagate through blood vessels.

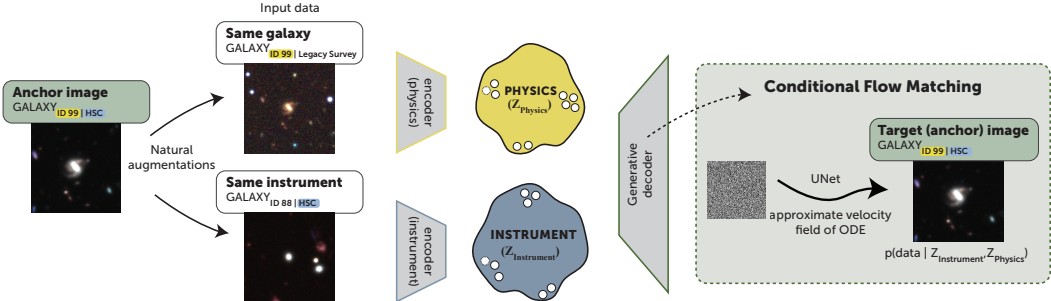

Figure 1: **Counterfactual Reconstruction Framework.** The model learns to disentangle intrinsic galaxy properties from instrument systematics by reconstructing an anchor image via a dual-encoder architecture and conditional flow matching. Training uses **data triplets** consisting of: an **anchor observation** (signal $s$ from instrument $i$), an instrument-augmented observation (**same source** $s$, a different instrument $i'$), and a physics-augmented observation (a different source $s'$, **same instrument** $i$). The *physics* encoder extracts latents from the instrument-augmented pair to capture invariant physical features, while the *instrument* encoder processes the physics-augmented observation to isolate measurement artifacts. These latent variables jointly condition the flow-matching decoder. Crucially, the anchor image is never fed into the encoders; it is used only as a target for the flow-matching loss, forcing the model to generate the reconstruction counterfactually.

This signal cannot be directly observed. Instead, it must pass through a sensor or instrument that records it. Any instrument used introduces measurement artifacts, often referred to as *instrument systematics* or bias. These effects include transformations like the nonlinearity of Charge-Coupled Devices (CCD) in a camera or a microphone's frequency response. These act as *confounding* factors. However, with sufficient understanding of the measurement system, they can often be partially modeled and disentangled from the underlying signal. Additionally, there is a contribution from noise and uncertainty. These are stochastic contributions that limit measurement precision. We treat this noise as stochastic residuals that are not explained by the latent variables learned from data.

In other words, *Observation* $= f(Signal, Instrument) + Noise$. Sometimes $f$ is simple, such as a straightforward point-spread function (PSF) convolution requiring only a good characterization of the optics. However, it often cannot be fully modeled analytically. The true signal is explained by variables that are independent of the measurement and contain the scientifically relevant information; we refer to these as **physics variables**. The instrumental effects arise from measurement distortions and act as confounders; we refer to these as **instrument variables**.

We present a general approach to learning to disentangle physics and instrument variables in settings where overlapping observations of the same physical system are available from different instruments. The key insight is that cross-matched observations form natural pairs: the same underlying physics observed under different instrument conditions. This provides a training signal for separating the two factors without requiring an explicit forward model of either instrument. We assume only black-box access to the instruments.

We demonstrate our approach using astrophysical galaxy images, a domain where modeling instrument systematics is critical. Photons emitted by a distant source must travel through cosmic dust and the Earth's atmosphere before being recorded by ground-based telescopes, each with its own unique optical system, detector characteristics, and observing conditions. As a result, the same galaxy observed by two different instruments can appear remarkably different due to differences in resolution, noise, and calibration. Historically, obtaining observations of the same phenomena with multiple instruments has been essential for distinguishing real physical signals from instrumental artifacts. For example, in the case of gravitational waves, the signal is buried deeply in detector noise, and coincident observations by multiple instruments are required to confirm detection.

In this work, we train our model on $\sim$100,000 cross-matched galaxy images from two different ground-based telescopes and show that we learn disentangled representations separating the intrinsic properties of galaxies from the instrument-specific effects, enabling robust inference of galaxy properties, counterfactual generation across instruments, and instrument-independent similarity search.

## 2 RELATED WORK

**Disentangled representation learning.** Learning representations that separate independent factors of variation is a longstanding goal in representation learning. Early work on variational autoencoders (VAEs), such as $\beta$-VAE (Higgins et al., 2017), encouraged disentanglement by regularizing the latent space via the loss function. A natural source of structure is multi-view (or multi-instrument) observations, where the same underlying content is observed under varying conditions or at different times. Denton & Birodkar (2017) proposed separate encoders for time-variant and time-invariant components in video, while Tran et al. (2017) used a dual-encoder architecture for pose-invariant face recognition. From a causal perspective, Schölkopf et al. (2021) framed disentanglement as the recovery of independent causal mechanisms, connecting representation learning and causal inference. Our framework draws on these ideas but enforces structural disentanglement through a specific architecture-driven information bottleneck and a counterfactual generation objective, rather than relying on hand-engineered loss functions.

**Foundation models in the Sciences.** Pre-training self-supervised representations on large scientific datasets has recently seen success in different areas of science like biology (e.g., Lin et al., 2023; Ross et al., 2022) and satellite imagery (e.g., Cong et al., 2022). In astrophysics, Walmsley et al. (2022) demonstrated the value of learned self-supervised representations for downstream tasks. More recently, AION-1 (Parker et al., 2025), developed a large-scale foundation model trained via masked modeling across different modalities such as imaging and spectra from the Multimodal Universe dataset (The Multimodal Universe Collaboration, 2024). However, a limitation common to these approaches is that instrument-specific effects are either ignored or treated as independent modalities without explicit disentanglement. This has practical consequences for downstream tasks. For instance, a model developed for ESA's Euclid mission (Euclid Collaboration et al., 2025) found that instrument systematics, rather than physically interesting sources, dominated the anomalies identified in the latent space. Our work addresses this by building the separation of physics and instrument factors into the architecture itself.

**Data-driven Noise Models.** As an independent and complementary approach to representation learning, several works have tackled modeling instrument effects through explicit data-driven modeling of noise and systematics (e.g., Legin et al., 2023; 2024). These require a detailed understanding of the noise structure and perfect separability from the physics. In contrast, our framework treats the instrument as a black box and learns its effects implicitly from cross-matched observations.

**Incorporating multi-instrument observations in Foundation Models.** Recently in astrophysics, Audenaert et al. (2025) leveraged the causal representation ideas from Schölkopf et al. (2016; 2021); Hattori et al. (2022), to causally separate astrophysical observations into a physics and instrument component by using a dual-encoder architecture. They used a dataset of simulated astronomical time series captured by multiple instruments to learn a *physics latent space* that encodes information about the target star and an *instrument latent space* that encodes information about the measurement configuration. This is achieved by leveraging triplets of overlapping observations and using contrastive learning. While we also leverage overlapping observations, our work steps away from contrastive learning in favor of a counterfactual generative modeling objective.

## 3 METHODS

We present a counterfactual generation objective designed to disentangle physical properties from measurement systematics within learned representations. Our goal is to capture robust physical information, shared across multiple measurements, into a physics latent space, while simultaneously learning a data-driven noise model for each of the instruments.

**Counterfactual Generative Objective.** Our framework leverages the growing availability of large-scale scientific datasets that contain overlapping observations from multiple instruments. In this setting, natural augmentations can be used to construct triplets: an anchor, observation of the same source (but different instrument), and observation on the same instrument (but different source), as shown in Fig. 1.

A *physics encoder* receives a sample from the target source but from a different instrument, so it is encouraged to reduce the loss by capturing the underlying physical properties of the source

and becoming invariant to the instrumental distortions. Similarly, an *instrument encoder* receives a sample from the target instrument but a different source, so it is encouraged to capture instrument-specific distortions while ignoring the content specific to the source. The decoder combines these two latent embeddings to reconstruct an unseen target or anchor observation.

Formally, consider a set of $N$ sources observed across $M$ instruments, where $x_{j,k}$ denotes the observation of source $j$ as captured by instrument $k$. We aim to learn the conditional distribution

$$p\big(x_{j,k} \,\big|\, \{z_{\text{phy}}(x_{j,k'})\}_{k' \neq k},\, \{z_{\text{ins}}(x_{j',k})\}_{j' \neq j}\big), \tag{1}$$

for all source–instrument pairs $(j, k)$. Here $\{z_{\text{phy}}(x_{j,k'})\}_{k' \neq k}$ are the physics embeddings extracted from observations of the *same source* $j$ by *other instruments* $k' \neq k$, which provide information about the source's intrinsic physical properties. The set $\{z_{\text{ins}}(x_{j',k})\}_{j' \neq j}$ are the instrument embeddings extracted from observations of *other sources* $j'$ by the *same instrument* $k$. This formulation allows the model to leverage all available multi-instrument data: the physics of the target source is informed by its appearances across different instruments, while the instrument response is informed by how other sources appear under similar observing conditions. Note that the number of source and instrument neighbors may vary; therefore, we combine the variable sequence of embeddings via attention-based conditioning.

We model this conditional distribution using flow matching (Lipman et al., 2023), learning a velocity field $u_\theta(x, t, z_{\text{phy}}, z_{\text{ins}})$ that transports samples from a standard Gaussian prior to the target distribution. The training loss is given by the error on the predicted velocity field,

$$\mathcal{L}_{\text{FM}}(\theta) = \mathbb{E}_{\substack{t \sim \mathcal{U}[0,1] \\ j \sim [N], k \sim [M] \\ x_{j,k} \sim p_{\text{data}} \\ x_0 \sim \mathcal{N}(0,I)}} \left[ \left\| u_\theta\left(x_t, t, \{z_{\text{phy}}(x_{j,k'})\}_{k' \neq k}, \{z_{\text{ins}}(x_{j',k})\}_{j' \neq j}\right) - (x_{j,k} - x_0) \right\|^2 \right] \tag{2}$$

where $x_t = (1 - t)x_0 + t x_{j,k}$. That is, during training, we sample a source $j$, a target instrument $k$, a noise sample $x_0$, and a timestep $t$, and train the velocity field to reconstruct the observation $x_{j,k}$ from an interpolated noisy state $x_t$ conditioned on physics embeddings from other instruments and instrument embeddings from different sources.

Once trained, we can use the learned representations for downstream tasks such as parameter inference, classification, or embedding-based retrieval, from the robust physics space, or find similar systematics and observing conditions from the instrument latent space. Interestingly, we can use the generative decoder to generate counterfactual views. A particularly promising use case in astrophysics is that of informing follow-up observations with more expensive instruments (e.g., James Webb Space Telescope) given cheaper low resolution observations from a different instrument.

**Comparison to contrastive methods.** While effective, contrastive approaches face several limitations that our generative framework addresses. First, contrastive objectives require carefully designed positive and negative pairs for each latent space, and the learned representations are sensitive to the composition of a training batch. Our attention-based conditioning accommodates a variable number of conditioning examples. Second, one would need to train a separate generative model on the learned embeddings to produce counterfactual views like the ones our framework produces, with no guarantee that the embeddings contain the pixel-level information needed to generate high-quality samples. In our framework, counterfactual generation is the training objective itself, so the representations are optimized to support it by construction. Finally, our approach implicitly learns a data-driven instrument model: the instrument encoder captures the dominant factors of variation in the observing conditions, while the flow-matching decoder translates these into pixel-level effects.

## 4 Application to Astrophysics: Galaxy foundation models

Datasets in astrophysics have grown considerably over the last decade. In particular, galaxy surveys such as DESI, have now imaged over a billion galaxies while taking spectra of over 10 million galaxies. This has sparked interest in developing foundation models for galaxies, such that we can learn semantic representations of astronomical data that can help us search large databases, identify interesting anomalies, and predict galaxy properties of interest. However, these foundation models have treated data from different instruments as independent modalities. Here, we demonstrate how

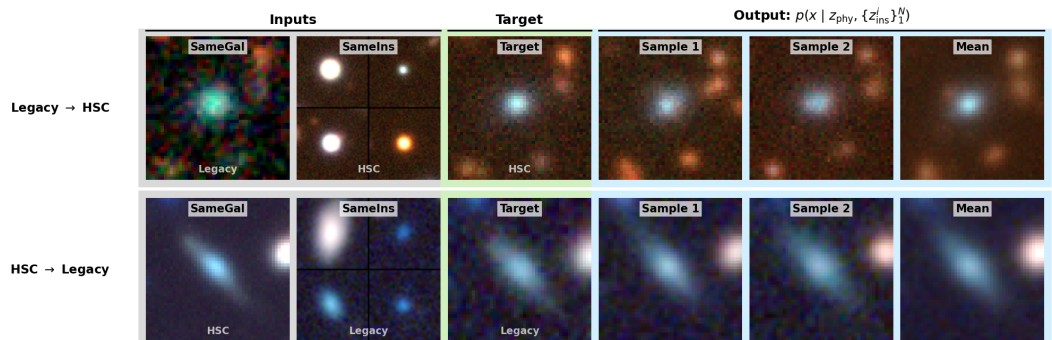

Figure 2: **Multi-instrument Galaxy Reconstructions. Columns 1–2 (Input Conditioning):** The target galaxy observed via an alternate instrument (input to the physics encoder) and a set of up to five different galaxies imaged by the target instrument (input to the instrument encoder). **Column 3 (Ground Truth):** The original anchor (target) image, withheld from the encoders. **Columns 4–5 (Posterior Samples):** Independent samples generated by the flow-matching decoder, demonstrating the model's ability to capture noise and uncertainty. **Column 6 (Empirical Mean):** The pixel-wise mean of the generated posterior samples, showing the recovered structural features of the galaxy.

our model can take advantage of the images from different instruments to extract robust physics information.

We take a cross-matched dataset from the Multimodal Universe that includes $\sim$100,000 cross-matched galaxy images from the DESI Legacy Imaging Surveys (Legacy, Dey et al., 2019) and the Hyper Suprime-Cam (HSC, Aihara et al., 2017) survey as a representative multi-instrument setting. HSC is a higher-resolution instrument that can detect significantly fainter features that would otherwise be masked by noise in the Legacy survey. However, by making this compromise legacy covers a much larger area of the sky, $\approx 20,000\,\mathrm{deg}^2$, compared to HSC, $\approx 1,200\,\mathrm{deg}^2$, resulting in a trade-off between coverage and signal-to-noise.

**Data Preprocessing.** For each entry in the dataset, we have two sets of heterogeneous flux measurements. HSC provides five color filters with discrete wavelength bands $(g, r, i, z, y)$, and the Legacy Survey uses four $(g, r, i, z)$. Despite the shared designations, they differ significantly in their central wavelengths, pixel scales, zero-points, and noise characteristics. Our preprocessing pipeline follows the strategy established by Parker et al. (2025), with one key difference: rather than treating all nine bands as distinct channels, we align the overlapping $(g, r, i, z)$ filters across both surveys into a unified four-channel representation and treat both sources as the same modality. For this purpose, we discard the additional HSC $y$-band channel, but future work could incorporate it.

To aid training stability, we normalize the zero-points by rescaling HSC measurements to the Legacy standard of 22.5 mag using the relation: $s = 10^{(ZP-22.5)/2.5}$. Then, we scale the flux values by the pixel-scale ratio to ensure spatial consistency and apply an arcsinh normalization to handle the images' large dynamic range. Finally, we crop the HSC images into $48 \times 48$ pixel cutouts. To account for the differing pixel scales between surveys ($0.168''$ for HSC vs. $0.262''$ for Legacy), we extract the inner $31 \times 31$ pixels of the Legacy images and upsample them to $48 \times 48$ pixels via linear interpolation; this ensures spatial alignment and a consistent field of view across pairs.

**Data Triplets.** In this dual-survey setting, each data point has a unique galaxy pair, providing the physics-augmented view. For the instrument views, we condition on up to five images of different galaxies observed by the same survey as the anchor. We select these as the five nearest spatial neighbors in the given survey within an angular separation of 3 arcminutes. This choice is motivated by the fact that instrumental systematics - such as PSF size and depth - vary smoothly across the sky, so spatially nearby observations share similar instrument conditions with the anchor. [1]

---

[1]We do not include images below the detection threshold of the source catalogs; incorporating sub-detection-threshold cutouts could improve instrument conditioning in future work.

**Model Implementation.** We use a ResNet-18 (He et al., 2015) as the backbone for both of our encoders (physics and instrument). We map each input image to a latent representation of shape $(B, 4, 16)$, where 4 denotes the flattened spatial dimensions and 16 is the embedding dimensionality. To accommodate a variable number of conditioning examples, we treat these embeddings as a set of tokens. For a given input set, the tokens from all images are concatenated into a single sequence of variable length. Then, we employ cross-attention layers within our velocity network, a `UNet2DConditionModel` (von Platen et al., 2022). This approach allows the model to leverage an arbitrary number of context image embeddings during the generative process.

## 4.1 RESULTS

**Counterfactual Generations.** Fig. 2 shows the model's ability to reconstruct target observations for both surveys and produce diverse posterior samples that capture the inherent uncertainty in the mapping. In addition to being our training objective, we can use counterfactual generation to prioritize follow-up observations. The discovery of rare objects, such as strong gravitational lenses, often depends critically on image quality. HSC is a deeper, higher-resolution instrument that allows us to resolve features, such as faint arcs, that are blurred or undetected in Legacy imaging. Our model can generate the expected HSC-like appearance for any galaxy in the Legacy footprint, effectively acting as a learned survey simulator that performs instrument-aware super-resolution and denoising. This could enable more targeted follow-up: one could first generate counterfactuals to identify the most promising objects, those where the model predicts lensing features that are plausibly present but partly unresolved in the Legacy data. The actual follow-up observation then provides the real photons needed for confirmation.

More broadly, this approach could support population-level studies by extending HSC-like morphological analysis across the full 20,000 deg² Legacy footprint. However, we caution that the model generates the statistically expected appearance given the Legacy input, not a true observation. It can sharpen and denoise features that are marginally detected, but it cannot recover information that is entirely absent from the input. Features well below the Legacy detection threshold will not appear in the counterfactual. The generated images are best understood as informed predictions that reduce the search space for follow-up, rather than replacements for real, deeper observations.

To evaluate the generated images, we report MSE computed on the generated images versus the preprocessed images (arcsinh-compressed and per-survey standardized pixel values). The posterior samples achieve an MSE of $0.081$ for HSC-anchored and $0.197$ for Legacy-anchored reconstructions, evaluated over 256 held-out galaxies.

In Fig. 3 (left), we further validate the uncertainty calibration of the flow-matching posterior. The pixel-wise Z-score, $Z = \frac{x - \mathbb{E}[\hat{x}]}{\text{std}(\hat{x})}$ where $\mathbb{E}[\hat{x}]$ and $\text{std}(\hat{x})$ are the posterior mean and standard deviation estimated from the generated samples, closely follows a standard normal distribution for both survey directions. However, we find that the model underestimates the true pixel-wise variance by approximately 15%. As a further validation, we test whether the generated HSC images preserve galaxy morphology. We train a ResNet18-based model to predict galaxy ellipticities from 25,000 real HSC images and apply it, without retraining, to counterfactual HSC images generated from Legacy images of held-out galaxies. Fig. 3 (right) shows that the predicted ellipticities match the ground truth comparably well for both real ($R^2 = 0.82$) and generated ($R^2 = 0.81$) images. Importantly, this means that an existing inference pipeline designed for HSC data can be used as-is on Legacy observations (by first passing them through our counterfactual model) yielding morphology measurements as accurate as those obtained from real HSC images and effectively extending HSC-quality morphological analysis across the entire Legacy footprint.

**Latent Space Visualization.** We encode 8,192 image pairs from HSC and Legacy Surveys using both the physics and instrument encoders, and visualize the resulting latent embeddings using a 2D UMAP projection (McInnes et al., 2020) in Fig. 4. The latent space demonstrates a clear separation of information. In the instrument space, HSC and Legacy observations form two distinct, non-overlapping clusters, confirming that the encoder successfully captures survey-specific characteristics. In the physics space, the two surveys have largely overlapping distributions, indicating that the encoder has learned an approximate domain-invariant representation of the underlying galaxy properties. The overlap is, however, not expected to be perfect: the two surveys provide different amounts of constraining power on the underlying physical properties. The physics encoder produces

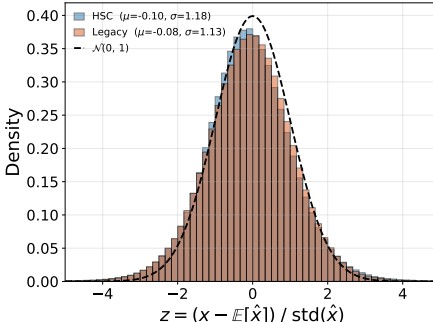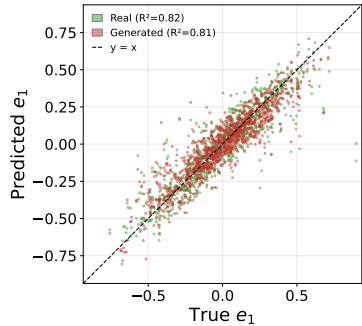

Figure 3: **Left:** Pixel-wise Z-score distribution of generated posterior samples relative to ground-truth target images, across all pixels and held-out galaxies. For each pixel, $x$ is the true value for a given pixel, $\hat{x}$ is the predicted one, $\mathbb{E}[\hat{x}]$ is the posterior sample mean over pixels and galaxies and $\mathrm{std}(\hat{x})$ is the posterior sample standard deviation. Both HSC-anchored and Legacy-anchored reconstructions closely approximate a standard normal distribution, indicating that the posterior mean is approximately unbiased, the posterior variance captures the true prediction error, and that the residual uncertainty is approximately Gaussian. The standard deviations slightly above unity indicate mild overconfidence, more pronounced for HSC-anchored reconstructions, consistent with the greater ambiguity in generating higher-resolution images from lower-resolution conditioning. **Right:** $R^2$ for galaxy ellipticity from a ResNet trained on real HSC images, evaluated on real versus counterfactual HSC images. Near-identical performance shows that existing HSC pipelines can be readily applied to Legacy images through our counterfactual model, extending HSC-quality analysis across the full Legacy footprint.

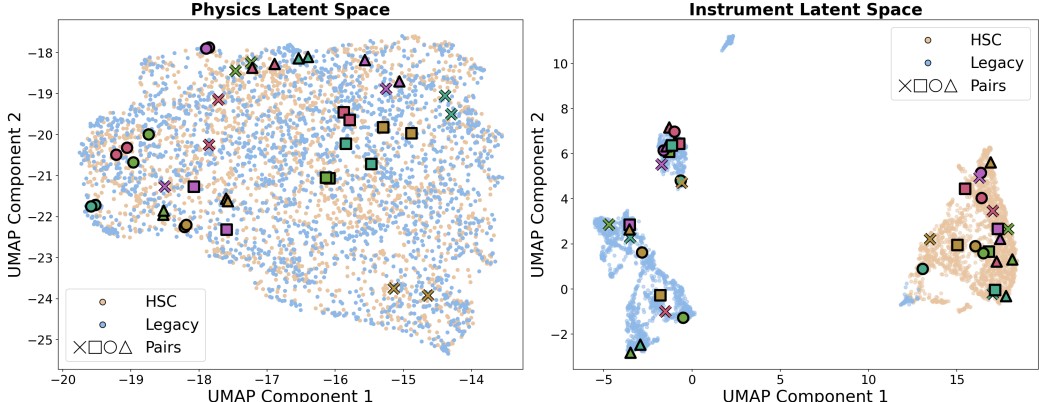

Figure 4: **Latent Space Disentanglement.** UMAP projections of the physics (left) and instrument (right) latent spaces. Orange and blue points represent HSC and Legacy images, respectively. Matched markers ($\triangle$, $\times$, $\square$, $\circ$) denote cross-survey pairs of the same galaxy. In the *physics space*, the encoders produce overlapping distributions where cross-survey pairs are mapped to similar coordinates. In contrast, the *instrument space* shows a clean separation between surveys and pairs, confirming that this space captures survey-specific features rather than intrinsic galaxy properties.

a point estimate of the posterior mean over physics latents, and deeper HSC imaging constrains galaxy properties more tightly than shallower Legacy imaging, resulting in a narrower distribution of posterior means.

Galaxy pairs (denoted by matching symbols in Fig. 4) map to nearby locations in the physics latent space, despite the training objective containing no contrastive component. This emergence of alignment of pairs from the reconstruction objective alone confirms that the architectural bottleneck is sufficient to drive the convergence of physics representations across the two surveys.

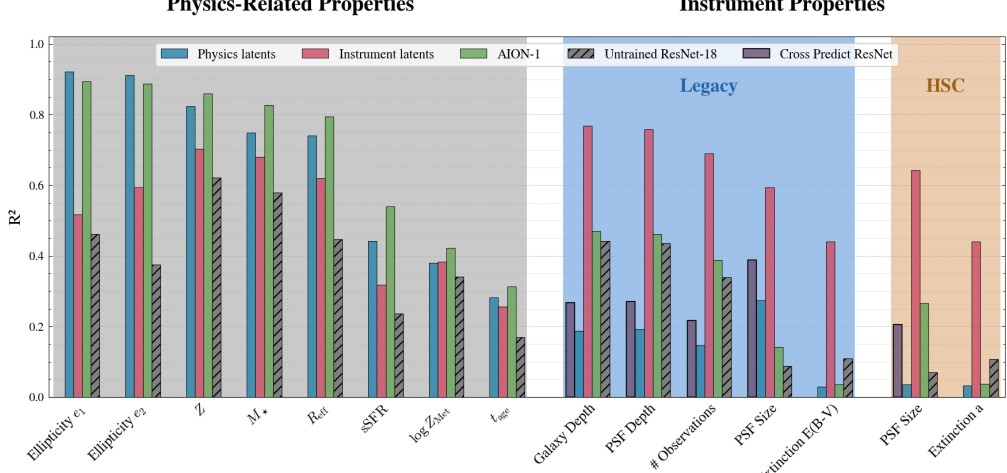

Figure 5: **Probing Latent Disentanglement via Downstream Regression.** We report $R^2$ scores for the prediction of physics-related properties (left) and instrumental properties (right) on four sets of representations: our physics encoder latents (blue), the instrument encoder latents (red), AION-1 embeddings which treat each survey as an independent modality (green), and a randomly initialized ResNet-18 of the same architecture as our encoders (hatched black). We distinguish physics-related properties (gray shading) and instrument properties, which are grouped by survey: Legacy (blue) and HSC (orange). For the instrument properties we add the performance of a ResNet based model trained directly to predict HSC instrument properties from Legacy images and viceversa (plum). The asymmetry between the two panels demonstrates that our architecture successfully separates physics-invariant information from instrument-dependent systematics.

**Downstream Tasks.** To evaluate the information content and disentanglement of our latent spaces, we train a small MLP to predict physical and instrumental properties from the frozen embeddings extracted by each encoder. We compare against the AION-1 (Base) embeddings, which treat the two surveys as independent modalities, and with a randomly initialized ResNet-18 of the same architecture as our encoders. Fig. 5 shows the resulting regression $R^2$ per property.

The physical properties of galaxies were inferred by fitting a physical model to the galaxy's stellar population in Hahn et al. (2023). However, we note that these are not "physics only" properties. For instance, the galaxy's redshift determines its apparent brightness, angular size, and observed colors - properties that are also changed by instrument conditions such as depth and PSF. Morphology is also affected by the instrument's PSF. Therefore, the instrument encoder will inevitably retain some of this information. On the other hand, observing conditions, such as PSF size and depth, vary across each survey's footprint as a function of sky position, determined by the tiling strategy and the atmospheric conditions during each field's observation. Therefore, the physics latent space may retain some of this information. We provide a detailed explanation of the different properties and the catalogue they came from in Appendix A.4.

In Fig. 5, to disentangle genuine leakage from shared spatial structure, we train a ResNet to predict each survey's instrument properties directly from the other survey's image ("Cross Predict ResNet"). This cross-prediction baseline measures the shared spatial component: any predictive power it achieves reflects information that is inherently common to both surveys and therefore accessible to either encoder by construction.

The physics latent space recovers physical properties at a level comparable to AION-1, despite it being trained to extract only information shared across HSC and Legacy surveys. Notably, the randomly initialized encoder already carries substantial information due to inductive biases of the ResNet architecture (Saxe et al., 2011). The instrument latent space retains some information about the physical properties, in particular redshift, stellar mass, and morphology. As explained above, this is to some degree expected, but we cannot quantify this. In the future, we will study how the

capacity of the instrument bottleneck affects these results, given that the untrained ResNet shows in some instances comparable predictive power.

The key diagnostic is the asymmetry in the split of properties: the physics latents substantially outperform instrument latents at predicting (mostly) physical properties, whereas the pattern reverses for instrument properties. Notably, the physics latent space contains less information about instrument properties than the cross-prediction baseline, indicating that it has actively erased instrument-related information beyond what is shared through sky position – one of the primary goals of our architecture. On the other hand, we note that AION-1 embeddings retain sensitivity to instrumental properties, but our instrument latent space outperforms both baselines in predicting instrument-related properties. This highlights the primary advantage of our architecture: by explicitly disentangling the latent factors, we achieve a more efficient separation of physics and instrumental properties than monolithic foundation models.

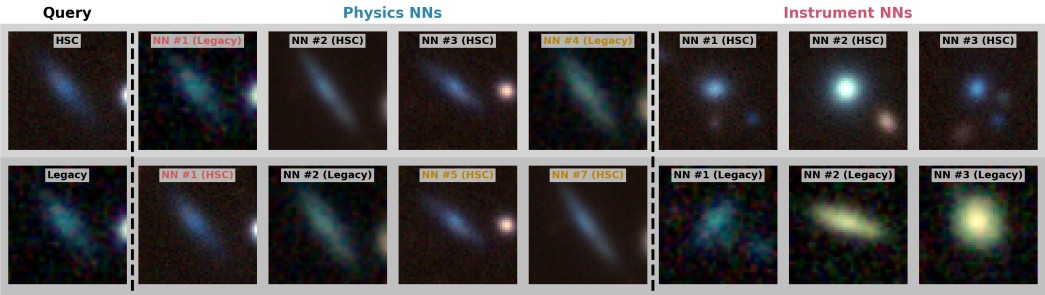

Figure 6: **Instrument-Invariant Nearest-Neighbor Search.** We evaluate the disentangled latent spaces by performing nearest neighbor retrieval using paired HSC and Legacy observations as queries. **Physics Space Retrieval:** For both queries, the corresponding pair from the alternate survey is identified as the top-1 neighbor (red). Additional top-ranked neighbors are physically similar galaxies from a mixture of both instruments (yellow indicates a cross-instrument NN). **Instrument Space Retrieval:** In contrast, neighbors retrieved in the instrument space show similar noise characteristics, independent of the underlying galactic signal.

**Neighbor search.** We can use the learned embeddings from our model to search for nearest neighbors (NNs) in the physics and instrument spaces. Often in astrophysics, we look for rare events such as galaxy mergers or strong lenses in large datasets that contain many millions of objects. Therefore, semantic embeddings that capture the broad characteristics of an image can be used to search for these rare events. Here, we demonstrate that searching the physics space enables us to find physically similar objects from either survey, whereas searching the instrument space enables us to find objects with similar noise properties. Fig. 6 shows a pair of HSC/Legacy observations that are used as queries. For each one, a mixture of HSC and Legacy images is found in the top neighbors in the physics space. In fact, the four examples shown could have been found among the top seven NNs, regardless of which survey the query came from. This demonstrates the ability to perform *instrument-independent similarity search* in physics space.

## 5 CONCLUSION

We have presented a framework for learning to disentangle instrumental effects from physical signals in scenarios with multiple views of the same object, i.e., observations from different instruments, using a simple counterfactual generative objective. Our dual-encoder architecture separates physics-invariant and instrument-specific factors structurally, without requiring hand-engineered contrastive losses or explicit instrument models. Compared to previous contrastive approaches, our method enables counterfactual generation by synthesizing how a source would appear under different observing conditions, and it is designed to scale to settings with more than two instruments. The generative component can also be viewed as a data-driven instrument model, learned implicitly from cross-matched observations. We have shown an application of our model to galaxy data in astrophysics, showing that the physics latent space recovers physical properties comparably to existing

foundation models while being robust to instrument systematics, and that counterfactual generation produces realistic cross-survey translations.

**Limitations and future directions.** Our approach currently requires overlapping observations across instruments, which limits applicability to regions of shared sky coverage. In future work, we aim to extend the framework to unpaired settings where such overlap is unavailable or limited. A second limitation concerns information asymmetry between instruments. Our current objective encourages the physics encoder to capture only the information that is shared across all instruments, discarding features that are unique to a single, potentially higher-quality instrument. For example, HSC's higher resolution reveals morphological details that are completely absent in Legacy imaging; our objective does not explicitly encourage preserving those in the latent space. A natural extension is to learn per-instrument residual latent spaces that capture instrument-specific physics information beyond the shared physics, complementing the robust shared representation with richer, instrument-dependent features. Finally, while we have focused on galaxy imaging, the framework is general and applicable to other domains in astrophysics and beyond. In particular, we plan to apply it to the tens of millions of light curves from NASA's Transiting Exoplanet Survey Satellite (TESS, Ricker et al., 2015) and Kepler (Borucki et al., 2010; Koch et al., 2010) missions.

ACKNOWLEDGMENTS

This work was supported in part by Advanced Micro Devices, Inc. under the AMD University Program's AI & HPC Cluster. The project that gave rise to these results received the support of a fellowship from "la Caixa" Foundation (ID 100010434). The fellowship code is B006068. We thank the AstroAI center at the Center for Astrophysics | Harvard & Smithsonian, where P M-P was a summer researcher under the supervision of DM, Rafael Martinez-Galarza, and Cecilia Garraffo, for their support and useful discussions on multimodal foundation models for astronomy. We also thank Michael J. Smith for helping access the dataset, and Rocco Di Tella and Rebeka Bottger for insightful technical discussions.

CODE AVAILABILITY

A simplified version of the codebase is publicly available.[2] For more information please contact `pablomer@mit.edu`.

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

# A APPENDIX

## A.1 MODEL ARCHITECTURE

We use a ResNet-18 (He et al., 2015) architecture for the encoders, plus a `UNet2DConditionModel` (von Platen et al., 2022) for the flow-matching generative decoder. The final pooling and projection layers of the ResNet are replaced with a convolution that produces a $2 \times 2$ feature map with 16 channels, which is treated as a sequence of 4 tokens of size 16 and passed to the UNet via attention conditioning (matching the cross-attention dimension of 16). Table 1 shows the architecture parameters used in the final training run.

Table 1: Model architecture parameters for the conditional flow matching model with dual ResNet-18 encoders.

| Parameter | Value |
|---|---|
| *Data Dimensions* | |
| Image size | $48 \times 48$ |
| Input channels | 4 |
| Conditioning channels | 4 |
| *UNet Architecture* | |
| Base model channels | 128 |
| Channel multiplier | $(1, 2, 4, 4)$ |
| Layers per block | 2 |
| Attention head dimension | 8 |
| Cross-attention dimension | 16 |
| Block types (down/up) | CrossAttnDownBlock2D / CrossAttnUpBlock2D |
| *Encoder Architecture* | |
| Backbone | ResNet-18 (timm) |
| Pretrained weights | False |
| Projection layer | Replaced by Conv2d($512 \rightarrow 16$, kernel=1) |

## A.2 TRAINING CONFIGURATION

The model was trained using distributed data parallel across 4 NVIDIA H100 GPUs with *bfloat16* mixed precision, completing 75,000 training steps in approximately 8 hours of wall-clock time. Table 2 shows the training configuration and hyperparameters used in the final training run.

## A.3 MODEL USED TO EVALUATE DOWNSTREAM PERFORMANCE

To evaluate the information content of our disentangled latent spaces, we perform supervised regression on a variety of physical and instrumental properties. We use a three-layer Multi-Layer Perceptron (MLP) with hidden dimensions of $[512, 256, 128]$, employing LayerNorm, GELU activations, and 20% dropout. The models are optimized using a Smooth L1 loss to ensure robustness against outliers in the catalog labels.

## A.4 GALAXY PROPERTIES USED FOR DOWNSTREAM TASK

**Dataset Composition.** Our evaluation is conducted on three distinct cross-matched subsets. For physical property (Z, Mstar, Reff, sSFR, logZMet, tage) regression, we use a sample of $4,278$ galaxies resulting from a cross-match between the PROVABGS spectroscopic catalog, the Legacy Survey and the HSC. This ensures that ground-truth spectroscopic labels are available for objects observed by both imaging surveys. We also use a sample of $4,096$ from the Multimodal Universe catalog to evaluate a mixture of physics-related and instrumental properties (Legacy extinction $E(B - V)$, HSC extinction coefficient 'a', ellipticity parameters e1, e2 and effective radius R). Finally, we draw a separate set of $4,096$ samples **from our own cross-matched neighbor search** results to evaluate localized observational metadata, including PSF characteristics (FWHM, shape), exposure counts

Table 2: Training configuration and hyperparameters.

| Parameter | Value |
|---|---|
| *Optimization* | |
| Optimizer | AdamW |
| Learning rate | $1 \times 10^{-4}$ |
| LR scheduler | CosineAnnealingLR |
| Loss function | Velocity MSE |
| *Flow Matching* | |
| Integration steps (sampling) | 250 |
| Interpolation | $x_t = (1-t)x_0 + tx_1$ |
| Target velocity | $v(x_t, t, c) = x_1 - x_0$ |
| *Training Details* | |
| Batch size | 64 |
| Total training steps | 75,000 |
| Number of devices | 4 (DDP) |
| Precision | bf16-mixed |
| Validation check interval | Every 1,000 steps |
| Train/val split ratio | 95% / 5% |

(Legacy $N_{OBS}$), and limiting magnitudes (Galaxy Depth, PSF Depth). See below a more in depth description of the different properies used.

**Physical properties.** Table 3 summarizes the physical parameters used to evaluate the physics latent space. These span a range of galaxy properties: morphological quantities (ellipticity and half-light radius) measured directly from HSC imaging, spectroscopic redshift from DESI, and stellar population properties (stellar mass, specific star formation rate, metallicity, and stellar age) derived from Bayesian SED fitting by PROVABGS (Hahn et al., 2023). The SED-derived quantities are inferred by jointly fitting stellar population synthesis models to DESI spectra and Legacy Survey photometry, and thus represent indirect measurements that integrate information beyond what is available in broadband imaging alone. Redshift is the most precisely determined of these quantities, as it is measured directly from spectral line positions.

Table 3: Physical parameters used for downstream regression evaluation of the physics latent space.

| Variable | Source | Description |
|---|---|---|
| Ellipticity $(e_1, e_2)$ | HSC catalog | Galaxy shape ellipticity components, measured via the SDSS-style second moments of the surface brightness distribution. |
| Redshift $(z)$ | DESI spectra | Spectroscopic redshift. |
| Stellar mass $(M_\star)$ | PROVABGS | Total mass in stars,, derived from Bayesian spectral energy distribution (SED) fitting to DESI spectra (Hahn et al., 2023). |
| Half-light radius $(R_{\text{eff}})$ | HSC catalog | Circularised radius enclosing half of the galaxy's total light. A measure of galaxy size. |
| sSFR | PROVABGS | Specific star formation rate $\log(\text{SFR}/M_\star)$: the rate at which new stars are forming normalised by the galaxy's stellar mass. |
| $\log Z_{\text{Met}}$ | PROVABGS | Gas-phase metallicity: the abundance of elements heavier than helium in the galaxy's gas. Traces the galaxy's chemical enrichment history. |
| $t_{\text{age}}$ | PROVABGS | Mass-weighted stellar population age: the average age of the stars in the galaxy. |

**Instrumental properties.**   Table 4 summarizes the instrumental and observational parameters used to evaluate the instrument latent space. These quantities describe the local observing conditions at each galaxy's position and vary spatially across each survey's footprint. For the Legacy Survey, the catalog provides direct measurements of depth (both for point sources and extended galaxies), PSF size, number of contributing exposures, and foreground dust extinction. HSC provides analogous quantities for PSF size and dust extinction, though computed differently as noted below. We evaluate the instrument latent space on properties from both surveys to assess whether the encoder captures instrument-specific conditions for the survey it is given, without encoding information from the other survey.

Table 4: Instrumental and observational parameters used for downstream regression evaluation of the instrument latent space. All quantities vary spatially across each survey's footprint.

| Variable | Bands | Description |
|---|---|---|
| *HSC (Hyper Suprime-Cam)* | | |
| `psf_fwhm` | $g, i, r, z$ | Effective size of the point spread function (arcsec): how much the instrument and atmosphere spread the light from a point source. Analogous to Legacy `PSFSIZE`.[†] |
| `a` | $g, i, r, z$ | Milky Way dust extinction $A_\lambda$ per band (mag). Equivalent to Legacy `EBV` scaled by a filter-dependent coefficient: $A_\lambda = R_\lambda\, E(B-V)$.[‡] |
| *Legacy Survey (DESI Legacy Imaging Surveys)* | | |
| `NOBS` | $g, i, r, z$ | Number of exposures contributing to the coadd. |
| `GALDEPTH` | $g, i, r, z$ | $5\sigma$ detection depth for a round exponential galaxy with $r_e = 0.45''$ (mag). Shallower than `PSFDEPTH` because galaxy flux is spread over more pixels. |
| `PSFDEPTH` | $g, i, r, z$ | $5\sigma$ detection depth for a point source (mag). |
| `PSFSIZE` | $g, i, r, z$ | Effective PSF size (arcsec). See HSC `psf_fwhm` above.[†] |
| `EBV` | — | Milky Way dust extinction as colour excess $E(B-V)$ (mag). See HSC `a` above.[‡] |

[†] Both quantities measure how much the instrument and atmosphere spread the light from a point source, but are computed differently: HSC derives it from the second moments of the coadd PSF model at each source position (Aihara et al., 2018); the Legacy Survey reports the exposure-weighted mean of the per-CCD PSF FWHM (Dey et al., 2019).

[‡] Both are derived from the same Milky Way dust maps. Galactic extinction is a foreground astrophysical effect that depends on sky position, not on the instrument; it is included here because it modifies the observed flux analogously to an instrument systematic.

## A.5   SPATIAL STRUCTURE PRESERVATION VIA SPECTRAL ANALYSIS

To test that the generated images preserve realistic spatial structure and correlation patterns, we compute power spectral density (PSD) and autocorrelation metrics across the reconstructed samples. For each direction (Legacy→HSC and HSC→Legacy), we randomly select a galaxy from a held-out test and generate 32 posterior samples using our model. We then compute the power spectral density as a function of spatial frequency and the autocorrelation as a function of pixel separation (lag), for each of the four imaging bands. This analysis shows that our generative model reproduces not just pixel-level statistics, but also the correlated spatial structures present in real astronomical images.

As shown in Fig. 7, both directions show strong agreement between ground truth and generated samples across different spatial scales, indicating that our model learns a realistic distribution over galaxy morphologies and instrument noise characteristics. The posterior uncertainty bands (shaded regions) appropriately expand at higher spatial frequencies, where noise dominates.

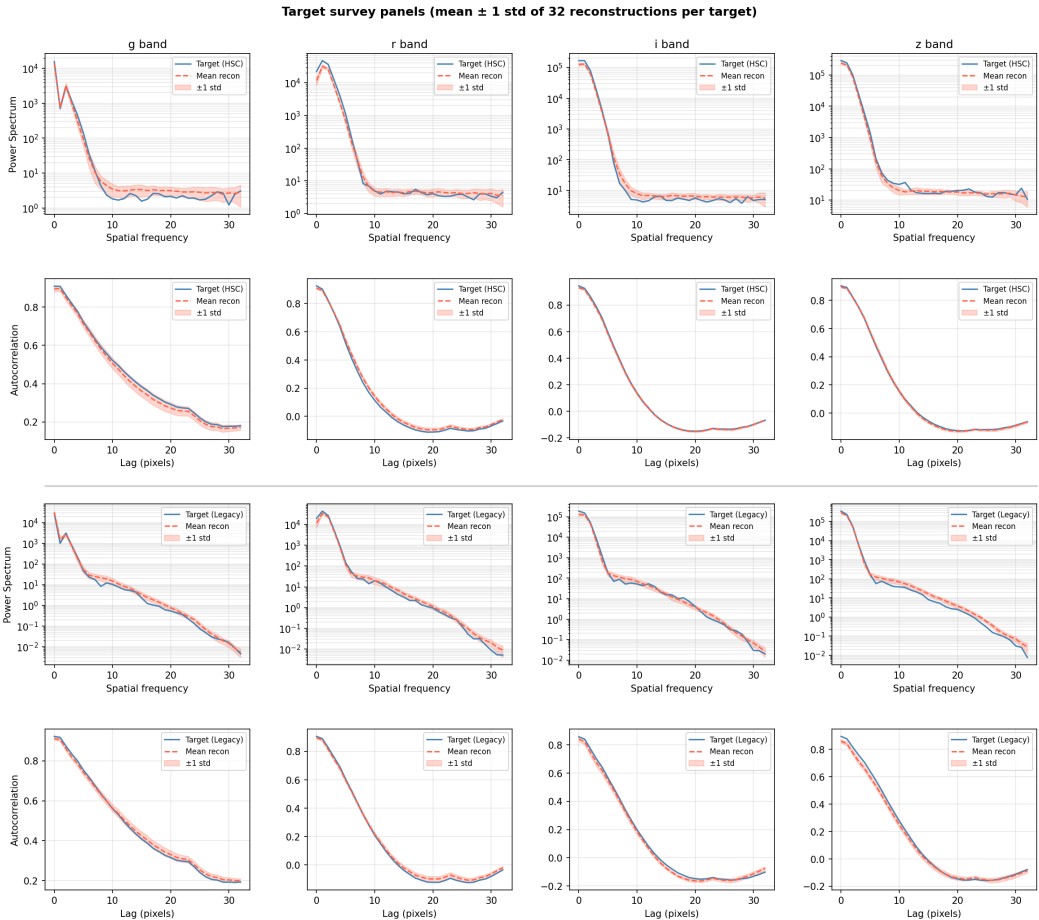

Figure 7: **Spatial Structure Preservation.** Power spectrum (left) and autocorrelation (right) per band for ground-truth and generated images. Each plot shows the true value (blue), the posterior mean from 32 generated samples (orange dashed line), and the $1\sigma$ posterior standard deviation (orange shaded region). Strong agreement across spatial frequencies and pixel separation (lag) distances indicates that the model captures realistic morphological structure and noise properties, rather than producing pixel-level artifacts, which would manifest as excess high-frequency power, or overly-smoothed outputs, which would appear as a deficit.

