# OpenReview forum: "Learning What's Real: Disentangling Signal and Measurement Artifacts in Multi-Sensor Data, with Applications to Astrophysics"
_ICLR.cc/2026/Workshop/FM4Science — ICLR 2026 Workshop FM4Science Poster_

### Official Review · Reviewer_kuPm · 2026-02-17
**The paper introduces a deep learning framework for disentangling intrinsic physical signals from instrument-specific artifacts in multi-sensor observational data. It leverages overlapping observations from different instruments as natural pairs, employing a dual-encoder architecture (physics + instrument encoders) and a counterfactual generation objective based on conditional flow matching. Applied to cross-matched galaxy images from the DESI Legacy Imaging Survey and Hyper Suprime-Cam Survey, the model produces invariant representations, enables counterfactual view generation (e.g., simulating higher-resolution images), supports robust parameter inference, and facilitates instrument-independent similarity search. The approach outperforms baselines in separating physics and systematics.**

**Rating:** 7
**Confidence:** 3

**Review:**

1. Quality:

Overall it is technically solid.  The architecture enforces factor separation structurally, i.e., dual encoders and conditional generator.  The learning objective directly trains the model for counterfactual reconstruction.  The experimental suite (generation examples, UMAP, regression probes, NN retrieval) is coherent and aligned with the claims.
Some main quality gaps exist. Disentanglement is demonstrated mostly via probes/visuals.  There is still measurable leakage, instrument latents retain some physics signal. The authors acknowledge confounding in “physics” labels, e.g., redshift affects observables similarly to depth/PSF.

2. Clarity:

The conceptual framing, e.g. physics vs instrument variables, counterfactual generation, is clear and well-motivated.  The conditional target distribution and flow-matching loss are stated explicitly.
Clarity could improve by adding more quantitative evaluation details in the main text, e.g., generation metrics, calibration of uncertainty, rather than relying primarily on qualitative figures and downstream probes.

3. Originality:

The key novelty is using counterfactual generative modeling (flow matching) as the primary disentanglement objective, rather than contrastive losses, while still leveraging overlapping multi-instrument observations.
The variable-length conditioning via attention is also a pragmatic design that naturally extends beyond two instruments.

4. Significance:

If it holds up broadly, this is significant for scientific self-supervised pretraining.  It offers a recipe to learn instrument-robust representations, while simultaneously producing a learned instrument-aware simulator useful for survey translation and follow-up prioritization.
The main limitation on impact is applicability. The current method assumes meaningful overlap across instruments, which unfortunately is not always available.

5. Pros:

* Structural disentanglement via dual encoders (physics vs instrument) without hand-crafted contrastive pairing.
* Counterfactual generation is the training objective, yielding both representations and a survey-translation model.
* Strong, task-aligned evaluations: UMAP separation, regression probes, NN retrieval across surveys.
* Uses a realistic large cross-matched dataset (~100k) with detailed preprocessing.

6. Cons:

* Requires overlapping observations across instruments, hence limited applicability where overlap is small or absent.
* Shared-only physics space may drop instrument-unique physical details (info asymmetry).
* Future residual physics spaces not yet validated.
* Evidence for generation quality is largely qualitative.
* Limited reporting of quantitative fidelity/uncertainty calibration.
* Leakage/confounding, i.e., instrument latents can still predict some physical properties, physics targets are not purely instrument-independent.

---

### Official Review · Reviewer_A84b · 2026-02-24
**A solid paper proposing a dual-encoder flow-matching framework for unsupervised physical and instrumental disentanglement in astronomy, and proved its effectiveness**

**Rating:** 9
**Confidence:** 4

**Review:**

1. Summary

Astronomy is a science based on observation, so from the very beggining of astronomical research, disentangling the intrinsic physical properties from instrument-specific observational effects is a fundamental task. This paper has given a delicate solution: to consider a dual-encoder architecture (physics + instrument decoders) while designing the conditional generative model based on flow-matching. Then, this paper has used several different ways to validate the effectiveness of these two encoders: this paper shows the latent space embeddings by UMAP projections; they also put the frozen embeddings to light MLP model to do regression for both physical and instrumental quantities; they also did similarity search with different encoder's embeddings. Overall, this is a well-executed paper with a rigorous logical loop, supported by both compelling qualitative visualizations and robust quantitative probing.

2. Strengths

- The design of forcing disentanglement through an architectural information bottleneck is highly clever. By feeding the Physics Encoder with cross-instrument views (same galaxy, different telescope) and the Instrument Encoder with cross-galaxy views (different galaxy, same telescope), the model cannot "cheat" by copying pixels. This naturally drives the latents into orthogonal semantic spaces. The UMAP projections (Figure 3) effectively visualize this, showing clear separation in the instrument space while merging cross-instrument counterparts in the physical space.

- The downstream regression probe (Figure 4) serves as the most convincing evidence of the method's success. By freezing the extracted embeddings and using them to predict both physical and instrumental properties, the authors demonstrate a clear asymmetry in R-squared scores. Notably, the physics embedding performs worse than a randomly initialized ResNet baseline when predicting instrument properties. This hard evidence proves that instrumental noise has been deliberately and thoroughly scrubbed from the physical latent space.



3. Weaknesses & Critiques

- While the nearest-neighbor retrieval (Figure 5) elegantly demonstrates semantic separation, the paper does not address the severe inherent statistical imbalance between the two latent spaces. In this paper, for a certain galaxy, its identical "physical neighbors" are incredibly rare (just "1" in this study), whereas "instrument neighbors" (galaxies sharing identical noise profiles/PSF from the same survey batch) are extremely dense and virtually infinite (although they only chose the 5 nearest). It is unclear how this massive density discrepancy affects the representation learning dynamics within the flow-matching framework, or whether the qualitative retrieval results are skewed by this inherent imbalance. Maybe an average of the 5 chosen galaxies, can be a better choice (just like the rightmost column in figure 2, seems the average is doable).

- Just like mentioned above, while choosing the instrument neighbors, they chose the 5 most nearest spatial neighbors. However, this may be possible that the nearest spatial neighbors would tend to share also similar physical properties due to their close position in the surverys. It seems to me that randomly chosen galaxies in the same instrumental survey, should be served as the "instrument neighbors".

---

### Official Review · Reviewer_FnEf · 2026-02-24
**Would need better evaluation on the generative tasks**

**Rating:** 6
**Confidence:** 5

**Review:**

This paper presents a thoughtful new framework for disentangling confounding effects given by the instruments from the data representing the physical phenomenon of interest. The paper is well motivated and in line with the workshop topics. The insight of using decoupled encoding from conditional counterfactual generation is novel, and interesting. The regression experiments in Figure 4 appropriately quantify cross-space predictability, demonstrating that the two encoders capture distinct information.

That said, the paper would benefit from a clearer organisation of the main goals and contributions. If the purpose is the sole disentanglement of features, achieving this with two separate encoders trained on different data is not really novel if not just logical. To me, the most interesting countribution comes from the foundation of this work as a solid bench for counterfactual conditional generation as per Eq. 1 and 2, which is however, not sufficiently tested nor demonstrated if not for the qualitative results in Figure 1.  I think the paper would benefit from stronger  baselines (e.g., a contrastive or domain-adversarial variant as opposed to the decoupled encoders) and evaluations of the generative task with quantitative metrics to evaluate cross-instrument reconstruction fidelity. Without these elements the paper feels unfinished, as most of the claims are demonstrated only for the representation learning task.
Finally, given the observation that a randomly initialized ResNet already carries substantial predictive power due to inductive biases, ablations with alternative backbone architectures would help clarify how much of the observed disentanglement arises from the training objective versus the encoder design itself.

More comments below:

- Figure 2 is not referenced in the paper

---

### Meta-Review · Area_Chair_UsrG · 2026-02-27

**Recommendation:** Accept (Poster)
**Confidence:** 3

**Metareview:**

Overall, the paper is solid, well-motivated, and aligned with the workshop theme. The dual-encoder architecture plus counterfactual flow-matching objective is coherent and methodologically sound, as also highlighted by the reviewers. However, some concerns remain about limited quantitative evaluation of the generative component, potential residual leakage between latent spaces, and applicability in settings without overlapping multi-instrument data. Despite these gaps, the paper is a meaningful and promising contribution.

---

### Decision · Program_Chairs · 2026-03-03

Accept (Poster)